# Dissecting the molecular organization of the translocon-associated protein complex

Stefan Pfeffer[1], Johanna Dudek[2], Miroslava Schaffer[1], Bobby G. Ng[3], Sahradha Albert[1], Jürgen M. Plitzko[1], Wolfgang Baumeister[1], Richard Zimmermann[2], Hudson H. Freeze[3], Benjamin D. Engel[1] & Friedrich Förster[1,4]

In eukaryotic cells, one-third of all proteins must be transported across or inserted into the endoplasmic reticulum (ER) membrane by the ER protein translocon. The translocon-associated protein (TRAP) complex is an integral component of the translocon, assisting the Sec61 protein-conducting channel by regulating signal sequence and transmembrane helix insertion in a substrate-dependent manner. Here we use cryo-electron tomography (CET) to study the structure of the native translocon in evolutionarily divergent organisms and disease-linked TRAP mutant fibroblasts from human patients. The structural differences detected by subtomogram analysis form a basis for dissecting the molecular organization of the TRAP complex. We assign positions to the four TRAP subunits within the complex, providing insights into their individual functions. The revealed molecular architecture of a central translocon component advances our understanding of membrane protein biogenesis and sheds light on the role of TRAP in human congenital disorders of glycosylation.

[1] Department of Molecular Structural Biology, Max-Planck Institute of Biochemistry, Am Klopferspitz 18, 82152 Martinsried, Germany. [2] Department of Medical Biochemistry and Molecular Biology, Saarland University, Building 44, 66421 Homburg, Germany. [3] Human Genetics Program, Sanford Burnham Prebys Medical Discovery Institute, 10901 North Torrey Pines Road, La Jolla, California 92037, USA. [4] Cryo-Electron Microscopy, Bijvoet Center for Biomolecular Research, Department of Chemistry, Utrecht University, Padualaan 8, 3584 CH Utrecht, The Netherlands. Correspondence and requests for materials should be addressed to B.D.E. (email: engelben@biochem.mpg.de) or to F.F. (email: f.g.forster@uu.nl).

Proteins synthesized on endoplasmic reticulum (ER) membrane-bound ribosomes must be either transported across or inserted into the ER membrane. These tasks are performed by the ER translocon[1], a multi-subunit membrane protein complex located in the ER membrane. The functional core of the translocon is formed by the universally conserved Sec61 protein-conducting channel, which is complemented by accessory translocon components, either assisting Sec61 or facilitating maturation of nascent chains by covalent modifications and chaperone-like functions[2]. One of these accessory translocon components is the translocon-associated protein (TRAP) complex[2–4], which was originally called the signal-sequence receptor (SSR) complex[5,6]. TRAP was found to be physically associated to Sec61 using biochemical methods[7–10] and has been chemically crosslinked to nascent proteins undergoing transport into the ER lumen[5,11,12]. TRAP was observed to stimulate translocation of proteins depending on the efficiency of their signal sequence in transport initiation[13]. Recent functional studies suggest that TRAP may affect the topology of transmembrane helices containing topogenic determinants that do not promote one specific orientation in the membrane[14]. Mutations in human TRAPδ (also known as SSR4) subunits were observed to result in a congenital disorder of glycosylation (SSR4-CDG), leaving some N-glycosylation sites unoccupied[15,16] and suggesting an additional role for TRAP in the biogenesis of N-glycosylated proteins.

Many inter-subunit interactions within the translocon appear to require an intact membrane environment, precluding detailed structural investigation of the assembled translocon complex in detergent-solubilized samples. Accordingly, single-particle cryo-electron microscopy (cryo-EM) studies of solubilized ribosome-bound translocon complexes have yielded fragmented and poorly resolved densities for the TRAP complex[17–20], likely due to disordered or partially dissociated TRAP subunits. In contrast, cryo-electron tomography (CET) in conjunction with subtomogram analysis can resolve the structures of membrane-embedded and -associated complexes within a natural membrane environment[21]. This technique enabled the first insights into the structure and molecular organization of the native translocon complex in rough ER vesicles derived from mammalian cells[18,22] and within undisturbed vitrified HeLa cells thinned by focused ion beam (FIB) milling[23]. In particular, these studies established

the positions of three major translocon constituents: Sec61, TRAP and the oligosaccharyl–transferase (OST) complex. Additional biochemically identified translocon components have not been successfully localized, likely because they are transiently recruited to the core components and therefore only present in a minor fraction of ribosome-bound translocon complexes[24]. Recently, technical advancements in CET made it possible to image the native ribosome-bound mammalian translocon at subnanometer resolution[25], revealing secondary structure elements for many parts of the translocon, including the TRAP complex.

By studying the native translocon in evolutionarily divergent organisms and disease-linked TRAP mutants, we detect structural differences that enable us to dissect the molecular organization of the TRAP complex. We assign positions to the four TRAP subunits in the assembled mammalian complex, providing new insights into membrane protein biogenesis and the role of TRAP in human congenital disorders of glycosylation.

## Results

**Structure and subunit composition of the human TRAP complex.** Mammalian TRAP is a heterotetrameric membrane protein complex[4]. The presence of classical cleavable signal sequences combined with bioinformatic analysis based on the positive inside rule[26] predicts that three TRAP subunits (α, β, δ) (SSR1, 2, 4) consist of an ER-lumenal domain (120–200 amino acid residues), one transmembrane helix and a small cytosolic domain (10–50 amino acid residues). TRAPγ (SSR3) is expected to contain a bundle of four transmembrane helices and a large cytosolic domain (100 amino acid residues) (Fig. 1a). Homology search (blast, psi-blast) in protein sequence databases reveals that the TRAP complex appears to be conserved in animals, but it is not uniform throughout the eukaryotic kingdom: plants and algae have a simplified subunit composition, lacking TRAPγ and TRAPδ, while TRAP is completely absent from most fungi. In the high-resolution subtomogram average of the ER-associated ribosome (EMD-3068)[25], mammalian TRAP is represented by three topological segments (Fig. 1b, Supplementary Movie 1): (I) a cytosolic domain bound to the ribosome via large subunit ribosomal RNA expansion segments (rRNA ES) and ribosomal protein L38, (II) a bundle of

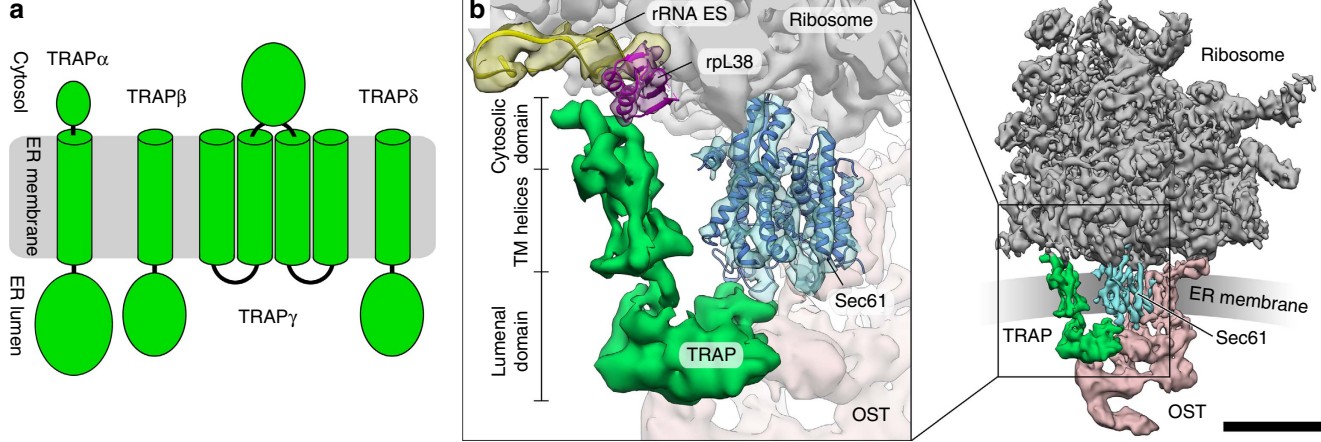

**Figure 1 | Overall structure and subunit composition of the mammalian TRAP complex. (a)** TRAP subunit composition and membrane topology as predicted by bioinformatic analysis. **(b)** Isolated densities for the mammalian ribosome (grey), the Sec61 protein-conducting channel (blue), TRAP (green) and OST (red) extracted from the high-resolution tomography density (EMD-3068) of the ER membrane-associated ribosome. In the magnified insert (left), atomic models for Sec61 (blue), rpL38 (magenta) and an rRNA ES (yellow) are superposed with the EM density to visualize interactions of the TRAP complex with the ribosome and the protein-conducting channel. Scale bar, 10 nm.

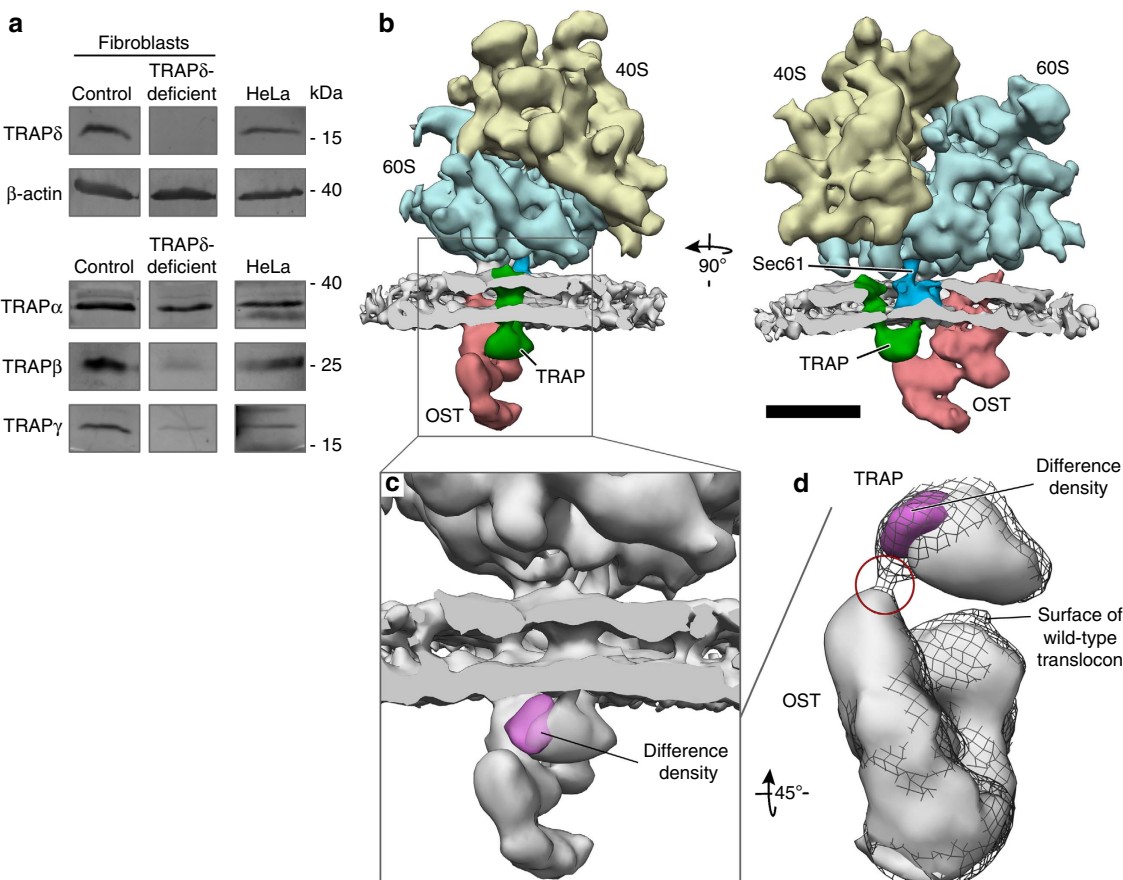

**Figure 2 | TRAPδ is located at the TRAP-OST interface. (a)** Western blot analysis of control and TRAPδ-deficient patient primary fibroblasts. HeLa cells were used as positive control for the TRAP antibodies. Please note that bands for HeLa cells and control fibroblasts are duplicated with Fig. 3a because samples were run on the same gel. **(b)** Subtomogram average of the ER-associated ribosome from TRAPδ-deficient primary fibroblasts, including the large (blue) and small (yellow) ribosomal subunits, the membrane bilayer (grey), the Sec61 protein-conducting channel (dark blue), OST (red) and TRAP (green). A total of 2319 subtomograms were averaged. Scale bar, 10 nm. **(c)** Zoomed view of the area indicated in **b**. The difference density map (magenta) between the TRAPδ-deficient and wild-type mammalian translocons is superimposed on the subtomogram average (grey). **(d)** Structure of the wild-type mammalian translocon (black mesh) superposed with the elements shown in **c**. TRAP and OST are shown from the perspective of the ER lumen, with the membrane removed. The contact site between TRAP and OST in the wild-type mammalian translocon is indicated (red circle).

transmembrane helices flanking the protein-conducting channel near the C-terminal half of the Sec61α subunit and, (III) an ER-lumenal segment, reaching across the Sec61 pore and binding to the ER-lumenal loop of the 'hinge' region between the N- and C-terminal halves of Sec61α.

**TRAPδ-deficient patient cells reveal the position of TRAPδ.** Recently, multiple mutations in human TRAPδ were observed to result in congenital disorders of glycosylation[15,16]. Examination of TRAP subunit abundance in patient primary fibroblasts and control fibroblasts by semi-quantitative western blot analysis indicated that this mutation leads to almost complete loss of TRAPδ, while the other three TRAP subunits are less severely reduced (Fig. 2a, Table 1). Most other tested ER-resident proteins and complexes were not significantly affected (Table 1), except for (I) an upregulation of the translocating chain-associated membrane protein (TRAM), which may have some functional overlap with TRAP[27] and could therefore partly counteract deficits in TRAP function, and (II) a downregulation of the nucleotide exchange factor Sil1, which may require TRAP for biogenesis.

To investigate how TRAPδ deficiency affects the structure of the TRAP complex, we isolated rough ER vesicles from patient primary fibroblasts and studied them using CET and subtomogram analysis. Consistent with earlier studies[18,25], computational sorting of automatically localized and iteratively aligned subtomograms depicting ER membrane-associated ribosomes yielded two translocon populations: one including (58%) and one lacking (42%) the OST complex (Supplementary Fig. 1a). Previous CET studies on mammalian cells characterized TRAP as a strictly stoichiometric component of the mammalian translocon present on all translocon complexes[18,23,25]. In contrast, further computational sorting of the OST-containing translocon complexes from the TRAPδ-deficient fibroblasts revealed a population of translocon complexes without density for TRAP (24% of the remaining subtomograms) (Supplementary Fig. 1a). This is consistent with the partial destabilization of all TRAP subunits in the TRAPδ-deficient fibroblasts observed by western blot analysis (Fig. 2a, Table 1). For the final average (Fig. 2b), only subtomograms depicting ribosomes bound to TRAP- and OST-containing translocons were retained, yielding a resolution of 15 Å (Supplementary Fig. 2a). The membrane bilayer and all translocon components (Sec61, OST and TRAP) were well resolved in the average.

We then compared the translocon density obtained for the TRAPδ-deficient fibroblasts to the structure of the wild-type mammalian translocon (EMD-3068). For an unbiased

**Table 1 | Relative protein content of TRAP subunits and select ER-resident proteins in patient fibroblasts.**

| Protein | TRAPγ-deficient fibroblasts | TRAPδ-deficient fibroblasts |
| --- | --- | --- |
| *TRAP complex* | | |
| TRAPα | 17 ± 4 (3) | 43 ± 1 (3) |
| TRAPβ | 15 ± 2 (2) | 17 ± 3 (2) |
| TRAPγ | 0 ± 0 (2) | 5 ± 3 (2) |
| TRAPδ | 12 ± 0 (2) | 2 ± 0 (2) |
| *ER membrane proteins* | | |
| Sec61α | 114 ± 17 (2) | 117 ± 22 (2) |
| Sec62 | 82 ± 10 (3) | 113 ± 9 (3) |
| TRAM | 141 ± 27 (2) | 205 ± 22 (3) |
| OST48 | 98 ± 0 (1) | 98 ± 0 (1) |
| RibI | 114 ± 0 (1) | 102 ± 0 (1) |
| *Lumenal proteins* | | |
| Sil1 | 45 ± 5 (4) | 72 ± 7 (4) |
| ERj3 | 70 ± 0 (1) | 104 ± 0 (1) |

Protein content given as % of control fibroblasts; normalized to β-actin; mean values ± s.e.m.; number of biological replicates shown in parenthesis.

comparison, both structures were filtered to 15 Å resolution, normalized according to the density's mean intensity value and standard deviation, and subtracted from each other to generate a density difference map. In the resulting difference map, exactly one highly localized and highly significant ($>7\sigma$) density reduction was observed for the TRAPδ-deficient translocon, defining the position of the TRAPδ subunit in the TRAP complex (Fig. 2c). The difference density (and thus TRAPδ) is located at the interface between TRAP and OST (Fig. 2d).

**TRAP is absent in TRAPγ-deficient patient cells.** Next, we analysed rough ER vesicles of fibroblasts from an unpublished mutation in human TRAPγ (SSR3), which was also observed to result in a congenital disorder of glycosylation. Examination of TRAP subunit abundance in patient primary fibroblasts and control fibroblasts by semi-quantitative western blot analysis suggested that the mutation leads to complete absence of TRAPγ (Fig. 3a, Table 1). However, levels of TRAPα were more severely reduced than for the previously analysed TRAPδ-deficient fibroblasts (Fig. 3a, Table 1), indicating that TRAPγ may be required for complex formation or integrity. Consistent with the western blot analysis of TRAPδ-deficient patient fibroblasts, most other tested ER-resident proteins and complexes were not significantly affected (Table 1), except for an upregulation of TRAM and a downregulation of Sil1 (see above).

Structural analysis of isolated rough ER vesicles from the TRAPγ-deficient primary fibroblasts using CET and subtomogram analysis yielded two translocon populations, one including (68%) and one lacking (32%) the OST complex (Supplementary Fig. 1b). Consistent with the low levels of all TRAP subunits observed by western blot analysis (Fig. 3a, Table 1), no density could be observed for the TRAP complex and even further computational sorting could not recover translocon populations with defined density for TRAP subcomplexes. Thus, for the final average (Fig. 3b), all subtomograms depicting ribosomes bound to the OST-containing translocon were retained, yielding a resolution of 22 Å (Supplementary Fig. 2b). The membrane bilayer and all translocon components except for TRAP were well resolved in the average.

We again compared the translocon density obtained for the TRAPγ-deficient fibroblasts with the wild-type mammalian translocon (EMD-3068) by computing a density difference map as described above. As expected from the total absence of

TRAP in the mutant structure, the density difference map showed a highly localized and highly significant ($>4\sigma$) reduction of density for the complete TRAP complex, including lumenal and cytosolic segments (Fig. 3c). In conclusion, structural analysis of the TRAPγ-deficient fibroblasts could not reveal the position of TRAPγ in the TRAP complex, but it confirms the importance of TRAPγ in stabilizing and coordinating TRAP within the ribosome–translocon complex.

**An algal translocon structure reveals the position of TRAPγ.** While TRAP is a heterotetrameric complex in mammals, it is composed of only TRAPα and TRAPβ in plants and algae. We decided to determine the structure of an algal translocon complex in the hope of confirming the position of TRAPδ and gaining insights into the positioning of TRAPγ. To resolve the structure of an algal translocon, we vitrified whole *Chlamydomonas reinhardtii* cells, which we then thinned with a focused ion beam[28,29] and imaged by CET[30,31]. A representative tomogram depicting a section of the native rough ER network within an undisturbed *C. reinhardtii* cell is shown in Fig. 4a.

Automatically localized and iteratively aligned subtomograms depicting ER membrane-associated ribosomes yielded a subtomogram average (Fig. 4b) at 19 Å resolution (Supplementary Fig. 2c). The lipid bilayer, Sec61 and TRAP were well resolved in the average. Initially no density could be discerned for the OST complex, suggesting that OST is highly underrepresented in the *C. reinhardtii* translocon. Consistently, computational sorting of subtomograms could recover only a minor population (14%) of OST-containing translocon complexes from the data (Supplementary Fig. 1c). Cell-type-dependent varying OST abundance has already been observed in mammalian systems analysed by CET, ranging from 35 to 70% (refs 18,23,25). As these algal ribosomes were imaged within their native cellular environment, we were also able to compare ribosomes bound to the ER to those attached to the nuclear envelope, and found that OST occupancy was similar for both populations. Notably, algal OST lacks a large ER-lumenal lobe compared to mammalian OST (Supplementary Fig. 1a,c), which yields a highly significant ($>5\sigma$) area of difference density attributed to the OST complex (Supplementary Fig. 3). Sequence alignments of lumenal segments for human and *C. reinhardtii* OST complex subunits show that only *C. reinhardtii* Ribophorin II diverges significantly from its human ortholog, lacking

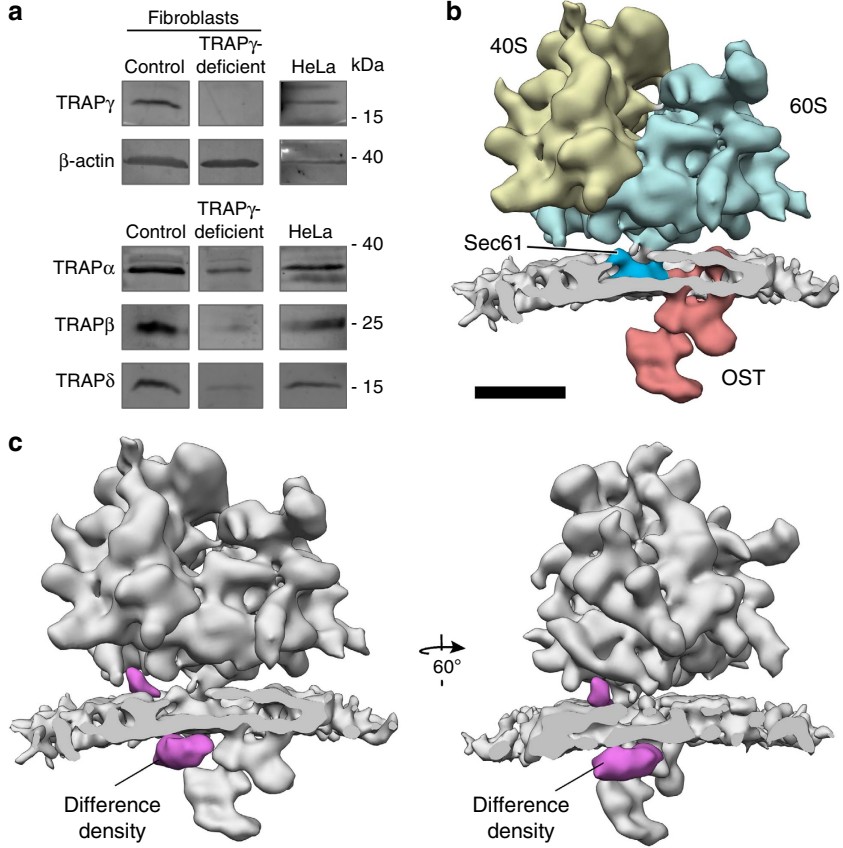

**Figure 3 | TRAP is destabilized in TRAPγ-deficient patient fibroblasts.** (**a**) Western blot analysis of control and TRAPγ-deficient patient primary fibroblasts. HeLa cells were used as positive control for the TRAP antibodies. Please note that bands for HeLa cells and control fibroblasts are duplicated with Fig. 2a because samples were run on the same gel. (**b**) Subtomogram average of the ER-associated ribosome from TRAPγ-deficient primary fibroblasts, including the large (blue) and small (yellow) ribosomal subunits, the membrane bilayer (grey), the Sec61 protein-conducting channel (dark blue) and OST (red). A total of 663 subtomograms were averaged. Scale bar, 10 nm. (**c**) The difference density map (magenta) between the TRAPγ-deficient and wild-type mammalian translocons is superimposed on the subtomogram average (grey).

a 30 kDa N-terminal domain that is predicted to project into the ER lumen. This suggests that the missing lobe in *C. reinhardtii* OST corresponds to the large N-terminal domain of mammalian Ribophorin II.

Because the low OST occupancy in *C. reinhardtii* limited the average of OST-containing translocon complexes to a resolution that was insufficient for reliable detection of differences in TRAP (30 Å), all subtomograms depicting ER membrane-associated ribosomes were used for the final average, irrespective of the presence or absence of OST. Accordingly, the heterogeneous OST region of the density was excluded from further analysis by masking. Comparing the algal translocon density with the wild-type mammalian translocon (EMD-3068) by computing a density difference map as described above, two highly localized and highly significant ($>7\sigma$) areas of density reduction could be attributed to *C. reinhardtii* TRAP (Fig. 4c). One of these areas (area 1) co-localized with the difference density obtained for the TRAPδ-deficient fibroblasts (Fig. 5a, Supplementary Movie 2), independently confirming the position determined for human TRAPδ. Thus, the first area of difference density in the algal translocon corresponds to TRAPδ. The second area of difference density (area 2) was located on the cytosolic face of the ER membrane (Fig. 4c) and co-localized with the cytosolic domain of mammalian TRAP that directly connects to the bundle of four transmembrane helices in the high-resolution tomography

density (Fig. 5a, Supplementary Movie 2). This arrangement corresponds precisely to the abovementioned predicted membrane topology of TRAPγ. Thus, the second area of difference density defines the position of the TRAPγ subunit in the mammalian translocon complex with high confidence.

## Discussion

Based on the set of difference densities obtained from the TRAPδ-deficient human translocon and the algal translocon (Fig. 5a, Supplementary Movie 2), the individual TRAP subunits can be assigned in the overall density of the mammalian TRAP complex (Fig. 5b). TRAPγ assumes a central position in the mammalian TRAP complex, binding to the ribosome on the cytosolic face of the ER membrane and coordinating the remaining TRAP subunits with the ribosome and the other translocon components. Thus, the ribosome-interacting function of TRAPγ may be required for TRAP occupancy at the translocon, explaining the complete absence of defined density for TRAP in the translocon of TRAPγ-deficient fibroblasts (Fig. 3b). The decreased stability of TRAP subunits in TRAPγ-deficient fibroblasts (Fig. 3a, Table 1) could furthermore hint at a central role of TRAPγ in assembly or stabilization of the mammalian TRAP complex. However, the ability of the TRAPα/β heterodimer to form a stable ribosome-bound subcomplex without TRAPγ in plants and algae argues against this function.

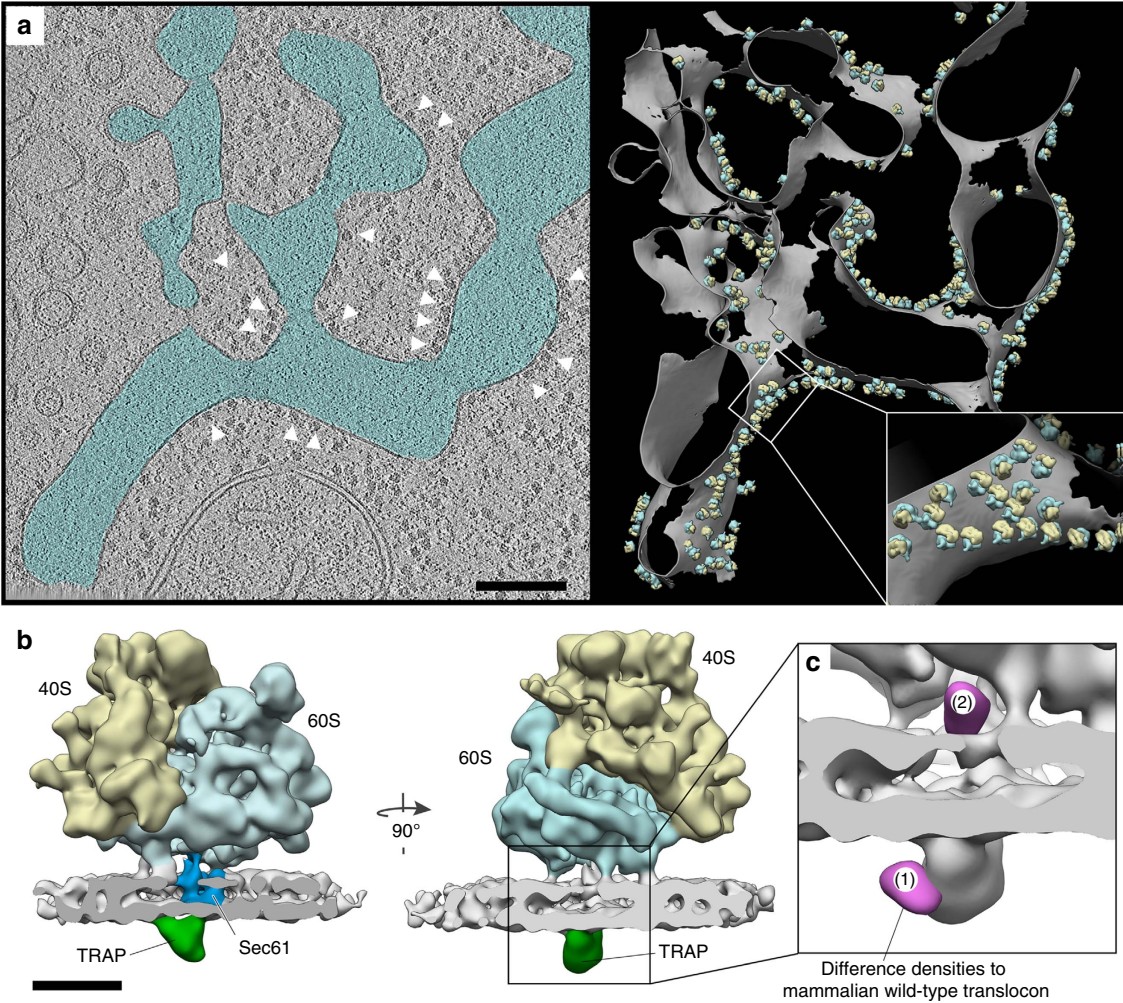

**Figure 4 | Structure of an algal translocon reveals the positions of TRAPγ and TRAPδ.** (**a**) Left: slice through a representative tomogram from a FIB-milled vitreous *C. reinhardtii* cell, depicting a section of the native rough ER network (blue). White arrowheads point to exemplary ER-associated ribosomes. Scale bar, 200 nm. Right: 3D rendering of the same tomogram, with the ER membrane depicted in grey and the small and large ribosomal subunits depicted in yellow and blue, respectively. (**b**) Subtomogram average of the ER-associated ribosome within native *C. reinhardtii* cellular volumes, including the large (blue) and small (yellow) ribosomal subunits, the membrane bilayer (grey), the Sec61 protein-conducting channel (dark blue) and TRAP (green). A total of 8070 subtomograms were averaged. Scale bar, 10 nm. (**c**) The difference density map (magenta) between the algal and wild-type mammalian translocons is superimposed on the subtomogram average (grey), revealing two localized areas of high density difference.

Thus, an alternative explanation for the decreased TRAP complex stability in TRAPγ-deficient fibroblasts is that it is not directly due to the lack of TRAPγ, but rather is caused indirectly by concomitant downregulation of TRAPα, assuming that TRAPα is the major complex stabilizing subunit.

The ER-lumenal domain of the TRAPα/β heterodimer contacts the loop of the hinge region between the N- and C-terminal halves of Sec61α and is positioned directly below the channel pore (Supplementary Movie 2). In this position, the TRAPα/β heterodimer could interact with translocating nascent polypeptides and influence the conformational state of the channel. This suggests that the TRAPα/β heterodimer (or only TRAPα in those few fungi that contain it) mediates the observed effects of TRAP on the topogenesis of membrane proteins[14] and the initiation of protein transport[13]. This is also supported by the later evolutionary gains of TRAPγ and TRAPδ in animals, which seem not to be strictly required for these processes. The absence of density for transmembrane helices of the TRAPα/β heterodimer (and TRAPδ) indicates that these transmembrane helices might not interact in a well-ordered fashion, while the

lumenal domains may mediate complex assembly. The cytosolic domains of these subunits are likely too small and flexible to be resolved, even in the higher-resolution subtomogram average.

TRAPδ is located at the periphery of the mammalian TRAP complex, which may explain the comparably mild effects of TRAPδ deficiency on the stability of the remaining TRAP subunits (Fig. 2a, Table 1) and their assembly into a stable TRAP subcomplex (Fig. 2b). TRAPδ is located at the interface between TRAP and OST (Fig. 2d, Supplementary Movie 2). Together with the observed congenital disorder of glycosylation upon loss of TRAPδ (refs 15,16), this suggests that TRAPδ plays a role in coordinating the functions of TRAP and OST in mammals. If cooperation between TRAP and OST is required for efficient co-translational *N*-glycosylation of certain mammalian proteins, the severe destabilization of the complete TRAP complex in TRAPγ-deficient fibroblasts (Fig. 3) would be sufficient to explain the observed congenital disorders of glycosylation. Interestingly, however, the substrate-dependent cooperation between TRAP and OST observed in mammals seems to not be required at all in the many other organisms that lack TRAPδ.

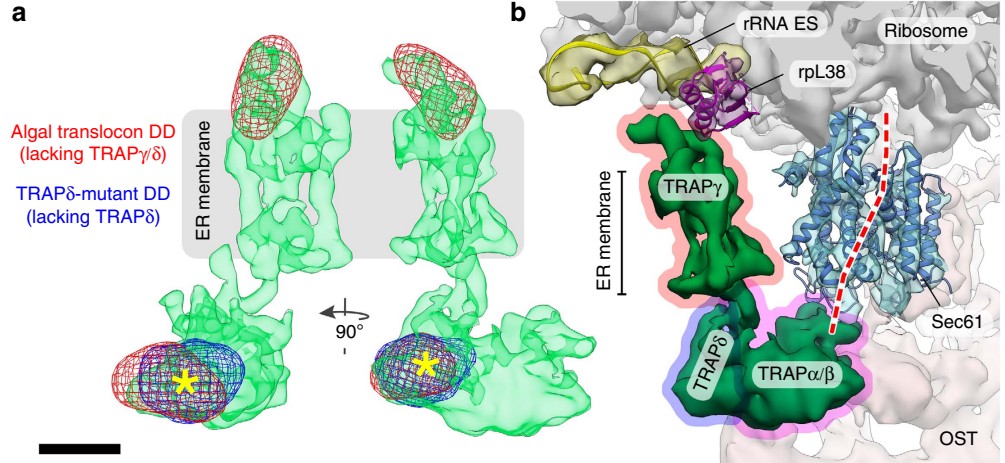

**Figure 5 | Dissecting the molecular organization of the TRAP complex. (a)** Set of difference densities (DD) originating from the TRAPδ-deficient fibroblast translocon (blue mesh) and the algal translocon (red mesh) mapped back on the isolated density of the TRAP complex from EMD-3068 (green). For the lumenal TRAP segment, difference densities originating from the TRAPδ-deficient fibroblast translocon and the algal translocon co-localize (asterisks). Scale bar, 2 nm. **(b)** Approximate positions of TRAPγ (red outline), TRAPδ (blue outline) and the TRAPα–β heterodimer (magenta outline), assigned according to the set of difference densities in **a**. The path of a nascent protein through the Sec61 complex (dashed red line) was traced based on PDB 5EUL. View and colouring matches Fig. 1b.

By determining the structure of the native translocon in evolutionarily divergent organisms and disease-linked TRAP mutant variants, we were able to assign positions to the four TRAP subunits in the assembled mammalian complex, providing insights into the roles the different subunits may play in membrane protein biogenesis. This refined view into the molecular organization of the TRAP complex provides a molecular basis for understanding the role of TRAP in human glycosylation disorders.

## Methods

**Cell culture.** *Human fibroblasts.* Primary fibroblasts from SSR4-CDG and SSR3-CDG patients were used with permission following informed consent. Dr Charles Marques Lourenço provided the cells from the SSR3-CDG patient. TRAPγ-deficient cells (CDG359), TRAPδ-deficient cells (CDG406) and control fibroblasts (GM0038, from the Coriell Institute) were cultured in DMEM 1 g per L glucose with L-glutamine and sodium pyruvate (HyClone, GE Healthcare, Freiburg, Germany) containing 10% FBS (Sigma-Aldrich, Taufkirchen, Germany) and 1% penicillin and streptomycin (Gibco, Thermo Fisher Scientific, Darmstadt, Germany) in a humidified environment with 5% $CO_2$ at 37 °C. None of the cell lines used in this study are listed in the database of commonly misidentified cell lines maintained by ICLAC. All cell lines have been tested for mycoplasma contamination. All cell lines used in this study have been either acquired directly from patients or obtained from the Coriell Institute, which maintains strict verification of cell lines.

*C. reinhardtii cells. C. reinhardtii mat3-4* cells (Chlamydomonas Resource Center, Univ. of Minnesota)[32] were cultured in Tris-acetate-phosphate (TAP) medium with constant light and aeration with normal atmosphere. Cells were harvested at early- to mid-log growth phase and diluted to ∼1,000 cells per μl in fresh TAP prior to vitrification.

**Preparation of microsomes from patient primary fibroblasts.** Cells (20–40 × 10⁶) were collected and successively washed with PBS and HEPES buffer (50 mM HEPES/KOH pH 7.5; 0.25 M sucrose; 50 mM KOAc; 6 mM MgOAc; 4 mM PMSF; 1 mM EDTA; 1 mM DTT; 0.1 mg ml⁻¹ cycloheximide; 0.3 U ml⁻¹ RNAsin (Promega, Heidelberg, Germany); protease inhibitor cocktail). Next, cells were homogenized in HEPES buffer with a glass/Teflon homogenizer, and the resulting lysate was cleared by two consecutive centrifugation steps (1,000g for 10 min and 10,000g for 10 min). The resulting supernatant was layered onto a 0.6 M sucrose cushion (50 mM HEPES/KOH pH 7.5; 0.6 M sucrose; 100 mM KOAc; 5 mM MgOAc; 4 mM DTT; 0.1 mg ml⁻¹ cycloheximide; 40 U ml⁻¹ RNAsin) to pellet membrane vesicles (230,000g for 90 min), which were subsequently resuspended in HEPES buffer and stored at −80 °C. All steps after the first washing step were carried out on ice.

**Semi-quantitative western blot analysis.** Western blots were scanned and analysed using the Typhoon-Trio imaging system (GE Healthcare) and the Image Quant TL software 7.0 (GE Healthcare). The primary antibodies were visualized using ECL Plex goat anti-rabbit IgG-Cy5 conjugate or ECL Plex goat antimouse IgG-Cy3 conjugate (GE Healthcare, Freiburg, Germany). Supplementary Table 1 lists primary antibodies used in this study. Full Western blot scans are shown in Supplementary Fig. 4.

**EM grid preparation and data acquisition.** *Rough microsomes.* Rough microsomes from patient fibroblasts were diluted using ribosome buffer (20 mM Hepes, pH 7.6; 50 mM KCl; 2 mM MgCl₂) and 3 μl were applied to lacey carbon molybdenum grids (Ted Pella, USA). After an incubation time of 60 s at 22 °C, 3 μl of 10-nm colloidal gold in ribosome buffer were added to the grid and the sample was vitrified in liquid ethane using a Vitrobot Mark IV (FEI Company, The Netherlands). Tilt series were acquired using a FEI Titan Krios transmission electron microscope (TEM) equipped with a K2 Summit direct electron detector (Gatan, USA), operated in movie mode with 4–7 frames per projection image (exposure time 0.8–1.4 s). The TEM was operated at an acceleration voltage of 300 kV, a nominal defocus of 3–4 μm and an object pixel size of 2.62 Å. Single-axis tilt series were recorded from −60° to +60° (first half: −20° to +60°; second half; −22° to −60°) with an angular increment of 2° and a cumulative electron dose of 90–100 electrons per Å² using the SerialEM acquisition software[33].

*C. reinhardtii cells.* Cells were vitrified by plunge-freezing with a Vitrobot Mark IV and thinned by focused ion beam milling with either an FEI Quanta or FEI Scios dual-beam microscope. After coating the samples with platinum, cells were thinned by scanning gallium ions from both sides in a stepwise fashion to produce final cellular sections that were 100–200 nm thick[28,34]. Milled samples were transferred to a 300 kV FEI Titan Krios TEM and imaged with a K2 Summit detector operated in movie mode at 12 frames per second. Using SerialEM[33], single-axis tilt series were acquired at 2° angular increments from approximately −60° to +60° (in two halves separated at either −20° or 0°), with an object pixel size of 3.42 Å and a cumulative electron dose of 70–120 electrons per Å².

**Tomogram reconstruction.** *Rough microsomes.* In-house developed frame alignment software based on the algorithm described in ref. 35 was used for correction of beam-induced motion in electron micrographs acquired with the K2 direct electron detector. Phase reversals resulting from the contrast transfer function were corrected in single projections with MATLAB and PyTom[36] using strip-based periodogram averaging[37]. Interactively located gold markers were used for tilt series alignment, and tomograms were reconstructed in PyTom[36] by weighted back projection (binned object pixel: 2.1 nm).

*C. reinhardtii cells.* Frames from the K2 direct electron detector were aligned as described above. Tilt series alignment by patch tracking, contrast transfer function correction of phase reversals and tomogram reconstruction by weighted back projection (binned object pixel: 1.37 nm) were performed in IMOD[38].

**Subtomogram analysis.** Candidate particles were located in the tomograms by template matching in PyTom[36] against single-particle cryo-EM reconstructions of human[39] (rough microsomes) or wheat germ[40] (*C. reinhardtii* cells) 80S ribosomes filtered to 50 Å resolution. For the 500 highest-scoring peaks of the cross correlation function, subtomograms were extracted from the tomograms and subsequently classified using constrained principal component analysis (CPCA)[41] focused on the ER membrane and the large ribosomal subunit. This allowed separation of ER membrane-associated ribosomes from non-membrane-bound ribosomes and most false-positive matches. The remaining subtomograms were reconstructed at full spatial sampling either with PyTom (rough microsomes: $220^3$ voxels, object pixel: 0.262 nm) or with IMOD (*C. reinhardtii* cells: $192^3$ voxels, object pixel: 0.342 nm) and iteratively aligned using PyTom[36]. Subsequently, CPCA focused on the translocon was used to separate ribosomes bound to the OST-containing translocon from ribosomes bound to the OST-lacking translocon, remaining non-membrane-bound ribosomes and false positives. For the TRAPδ-deficient fibroblasts, a third round of CPCA focused on the TRAP complex separated ribosomes bound to the TRAP-containing translocon from ribosomes bound to the TRAP-lacking translocon. The number of subtomograms used for averaging in each dataset is noted in Supplementary Fig. 1. For all data sets, Fourier shell correlation of two maps derived from each half of the data (FSC = 0.5) and Fourier shell cross resolution (FSC = 0.33) against a cryo-EM single particle reconstruction of the human (rough microsomes)[39] or wheat germ (*C. reinhardtii* cells)[40] 80S ribosome were used to assess the resolution of the subtomogram averages.

**Analysis of EM densities.** The UCSF Chimera software package[42] was used for EM map analysis, fitting of atomic models, segmentation and visualization. For computation of difference density maps, the two EM densities were filtered to the same resolution (TOM/av3), aligned onto each other (Chimera), interpolated on a common reference system (Chimera), normalized according to the density mean and density standard deviation (TOM/av3), and subtracted from each other (TOM/av3). Due to the heterogeneous OST occupancy on the ER membrane-associated ribosome from *C. reinhardtii* cells, the OST region was masked out from both EM densities before computing the difference density, in this case. Segmentation of the ER membrane in Fig. 4a was performed in Amira software (FEI Visualization Sciences Group), guided by a tensor voting algorithm[43].

**Data availability.** Subtomogram averages of the ER membrane-associated ribosome from human TRAPδ- or TRAPγ-deficient fibroblasts and *C. reinhardtii* cells have been deposited in the EMDataBank with accession codes EMD-4143, EMD-4144 and EMD-4145, respectively. All the remaining data are available from the corresponding authors upon reasonable request. Previously published electron microscopy densities (EMD-3068, EMD-5592, EMD-1780) and atomic models (PDB 5EUL) were used in this study.

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

## Acknowledgements

We are grateful to Monika Lerner (Homburg) for excellent technical assistance. This work was supported by grants from the Deutsche Forschungsgemeinschaft to F.F. (FO 716/4-1) and R.Z. (ZI 234/13-1). H.H.F. and B.G.N. are supported by The Rocket Fund and R01DK099551.

## Author contributions

M.S. performed FIB milling and, together with B.D.E. and S.A., acquired tomography data from *C. reinhardtii* cells. B.G.N. identified SSR4-CDG and SSR3-CDG patients and collected patient fibroblasts. J.D. performed patient fibroblast culture, preparation of rough microsomes and western blot analysis. S.P. acquired tomography data from patient fibroblast microsomes and, together with B.D.E. and S.A., carried out analysis of CET data. J.M.P., W.B., R.Z., H.H.F., B.D.E. and F.F. planned and supervised the experiments and, together with S.P., J.D. and B.G.N., wrote the manuscript.

## Additional information

**Competing financial interests:** The authors declare no competing financial interests.

