## [Peer review file · Nature Communications]

Reviewers' comments:

Reviewer #1 (Remarks to the Author):

The manuscript by Pfeffer et al. combines a deeper analysis of the data published in 2015 in Nat. Comm. with new experiments in order to define the position of TRAP subunits in the translocon. Doing so, they are able to identify possible positions of parts of the TRAP δ and TRAP γ subunit. The so identified position of a TRAP δ domain in proximity to the OST could be an explanation for the observed underglycosylation phenotype in patients with TRAP δ mutations.

All these findings are new they are new, but only of limited general interest.

General question:

With their method, the authors nicely identify the Sec61 complex, the OST and the TRAP complex in the native translocon. However, they seem to miss other well established constituents, like the TRAM protein or the Sec63. The authors should mention and if possible comment this problem in the discussion.

Minor points

The predicted orientation of the single-spanning TRAP components are more easily explained by the presence of a cleavable signal sequence. Moreover, there are biochemical data that support this prediction (e.g. the inhibition of protein transport by anti-TRAP α antibodies directed against the c-terminus of the protein)

Page 6 – at least according to the data presented in table 1 TRAP β is not more severely reduced in the TRAP γ deficient fibroblasts compared to the TRAP δ deficient ones.

Reviewer #2 (Remarks to the Author):

Pfeffer et al. have carried out a cryo-electron tomographic (CET) analysis of the organization of the translocon-associated protein (TRAP) complex. The mammalian TRAP complex is comprised of 4 subunits, whereas in plants and algae there are only two subunits. Furthermore, the TRAP gamma subunit is missing, or present in low amounts, in cells from patients with congenital disorders of glycosylation. Difference-CET analysis of the ribosome/Sec61/OST/TRAP complexes from these different systems allowed Pfeffer et al. to determine the locations of all four of the subunits of the normal mammalian TRAP complex. Their findings, summarized in Figure 5, reveal the general architecture of the ribosome/Sec61/TRAP complex. TRAP-gamma with four putative transmembrane helices, Sec61, and the lipid bilayer are clearly revealed in the tomograms. Furthermore, the tomograms indicate that TRAP-gamma is connected to the ribosome and TRAP-delta, which allows TRAP-alpha/beta to connect to Sec61 and oligosaccharyltransferase (OST).

This is important and exciting work. Although TRAP was apparent in earlier cryo-EM studies of the mammalian ribosome/Sec61 complex, its structural organization was not apparent. The present work reveals clearly for the first time the TRAP structural organization and its spatial relationship to the ribosome, Sec61, and OST. Besides being informative, I believe the work will stimulate structural studies of the TRAP subunits and thereby provide new insights to membrane protein biogenesis. A very minor point is that the meaning of OST is not clearly spelled out in the manuscript text.

Reviewer #3 (Remarks to the Author):

Manuscript Summary / Overview

In this manuscript the authors apply state-of-the-art cryo-ET imaging, subtomogram averaging, and computational sorting procedures to study translocon structure and function. By comparing algal and mutant human ribosome / translocon complexes with a previously published solution for the wild-type mammalian structure, they are able to assign presumptive locations to the TRAP delta and gamma (and by exclusion, alpha and beta) subunits. Combining these locations with information on the absence of specific subunits in some kingdoms permits inferences about the functions and mechanisms of action of the various subunits.

Key experimental observations are:

1. Bioinformatic analysis suggests that TRAP alpha, beta, and delta should contain small cytosolic domains, single transmembrane helices, and a sizeable ER luminal domain. In contrast, TRAP gamma is expected to have 4 transmembrane helices and a large cytosolic domain.
2. Homology searches indicate that TRAP gamma and delta are absent in plants and algae, while most fungi lack TRAP altogether.
3. Difference maps between human fibroblast TRAP delta- mutants and a previous wild-type mammalian structure show a single, compact region of high significance near the interface between TRAP and oligosaccharyl transferase (OST). Since Western blotting shows nearly complete elimination of TRAP delta (2% of wild type), while the other subunits are less impacted (43, 17, and 5% of wild type, respectively for alpha, beta, and gamma), the authors conclude that this region indicates the location of TRAP delta.
4. A similar attempt was made to localize TRAP gamma using a gamma- mutant. Western blotting indicated complete elimination of gamma in the mutant, but also substantial

reduction of alpha, beta and delta to 17, 15, and 12% of wild type levels. Unfortunately, no TRAP complexes were found in translocons from gamma- mutants.

5. Western blots from the delta- and gamma- mutant strains indicate nearly complete elimination of the corresponding protein, but also substantial reductions in other TRAP subunits and a couple additional proteins.

6. Since algal TRAP contains alpha and beta but neither gamma nor delta subunits, the authors computed a difference map between the mammalian and *Chlamydomonas reinhardtii* complexes in another attempt to localize the gamma subunit.

a. OST was not well-represented in the *Chlamydomonas* samples (~14% of samples). Analysis of complexes that do contain OST indicated that algal OST lacks a large ER luminal domain found in mammals. Sequence analysis suggests that the missing region corresponds to an ~30 kDa N-terminal region of human Ribophorin II.

b. Due to the scarcity of OST containing complexes in *Chlamydomonas*, difference mapping between algal and human complexes was conducted while masking out OST. Two regions of highly significant differences were observed, one corresponding to the location previously assigned to TRAP delta. The other region is located on the cytosolic face of the endoplasmic reticulum in a position corresponding to the interface between the cytosolic domain of mammalian TRAP and 4 transmembrane helices. This is consistent with the predicted structure for TRAP gamma, so TRAP gamma is assigned to this position, spanning the membrane.

7. By a process of elimination, TRAP alpha and beta are assigned to the remaining TRAP density, a relatively compact region just below the Sec61 channel.

Based on these observations, the authors propose that:

1. In animals, TRAP gamma is essential for stability of the TRAP complex and for coordination with the ribosome / translocon complex. Gamma and delta are neither present nor required

in plant and algal TRAP, so authors suggest that this need for stabilization by gamma in animals may be a relatively late evolutionary addition.

2. The location of the TRAP alpha / beta dimer just below the Sec61 channel makes it well positioned to interact with translocating peptides and to control channel conformation.

3. The delta subunit is located at the periphery of the TRAP complex, perhaps explaining the relatively minor impact of TRAP delta- mutants on stability compared to gamma- mutants. Both TRAP delta- and gamma- mutants are associated with clinically significant, congenital disorders of glycosylation, however, suggesting that delta plays an important role in coordinating TRAP and OST in mammals.

Overall Recommendation

Understanding protein processing by the translocon and the endoplasmic reticulum is an important topic of broad general interest. To the best of my knowledge, the results presented here are novel and significant, the methods used appropriate, and very well chosen and applied. Indeed, ongoing results from this group, including this manuscript, are excellent examples of how combining subtomogram averaging with computational techniques can provide insight into biologically relevant structures too large or too unstable when isolated for single particle analysis. Overall, the conclusions drawn are reasonable and well-supported by the data. Indeed, they may well be the simplest, most likely explanation for the observed results. Nevertheless, it seems to me that an alternate explanation may be possible in at least one important case, described below. Therefore, I recommend that this article be accepted for publication after editorial revision. Specifically, I suggest that the authors consider adding further discussion of alternative explanations should they and the editors agree these concerns are reasonable.

Comments, Questions, and Suggestions

In the following, I will begin with rather trivial typographical errors and formatting suggestions, proceeding to more substantive concerns.

1. Typo on page 8. Should say “wild-type” rather than “wilt-type”.

2. Typo on page 11. Should say “Human fibroblasts” rather than “Human fibtoblasts”.

3. Oligosaccharyl transferase should probably be fully spelled out when first introducing the OST abbreviation on page 5.

4. In Figures 2 and 3, the “transparent red” used to flag the significantly different regions is easily confused with red color used as a key for OST in other parts of the figure. Changing either to a different color would make the figure more readily understandable.

5. Lines and / or a box might be useful to indicate the relation of Figure 2d to 2c, as was previously done for 2b and 2c. While the 45 degree rotation is clearly indicated, the change in scale was not immediately apparent, and it took me a moment to orient myself and realize what was being shown.

6. Also in Figure 2, why does the TRAP / OST interface show up clearly in the wild-type translocon isosurface but not as a region of significant differences? If TRAP delta is the major contributor to interaction with OST, I'd expect this region to also be missing or greatly attenuated in the TRAP delta- mutants. Is this simply a matter of the high, seven standard deviation, threshold chosen for the difference map?

7. Page 10 of the Discussion reads, in part, “The absence of density for transmembrane helices of the TRAP alpha/beta heterodimer (and TRAP delta) indicates that their transmembrane helices might not interact in a well-ordered fashion...”. Is it worth also commenting on and explaining the absence of detected cytosolic regions for these proteins, especially in the case of alpha which, according to the bioinformatics analysis, should have the largest of the “small” cytosolic domains?

8. Why does no TRAP gamma significant difference show up in the lumen? Is this region simply too small?

9. Page 10 of the Discussion has a parenthetical statement reading “(or only TRAP alpha in some fungi)”. Given that in the Results it was previously stated that TRAP is completely missing from most fungi, would it perhaps be more clear if this parenthetical statement was changed to something like “(or only TRAP alpha in those few fungi which contain it)”?

10. In both Figures 1a and (presumably) in 5a, gray shading is used to indicate the membrane region, but this is never explicitly stated. It's pretty apparent that this is what's meant in 1a, but perhaps less so in 5a. It might also be worth indicating the transmembrane region once again in 5b (although not with gray shading!). This was already done previously in 1b, so it's doesn't seem strictly necessary, but might help to indicate relative scale and positioning between 5a and 5b.

The authors use Western blots to confirm protein abundance of the mutant strains microsomal fraction relative to wild type. As expected and hoped, both delta- and gamma- strains show almost complete elimination of the corresponding protein. Other TRAP and ER proteins are also affected, as indicated in Table 1. This is not surprising. Biological regulatory and functional networks are notoriously promiscuous, and feedback loops, co-regulation, and pleiotropic behavior are common in addition to effects on stability. This makes separating cause from effect challenging, leading to several questions and concerns.

11. TRAM appears to be significantly up-regulated (259% of wild type) in the delta- mutant, while Sil1 is significantly down-regulated (48% of wild type) in gamma-. Do the authors have any comments on these changes?

12. The statement on page 6 that “levels of TRAP alpha and TRAP beta were more severely reduced than for the previously analyzed TRAP delta-deficient fibroblasts (Fig. 3a, Table 1)” in the TRAP gamma- mutant, while literally true, seems over-stated and somewhat misleading. It unquestionably applies to alpha which is reduced to $17 \pm 4\%$ of wild type levels in the gamma-deficient mutant versus $43 \pm 1\%$ in delta-. For beta, however, the corresponding levels of $15 \pm 2\%$ and $17 \pm 3\%$ (both with N of only 2) are virtually identical.

13. My greatest concern is with the observation that TRAP alpha is reduced much more dramatically in gamma- than in delta-. Is this an effect of reduced TRAP stability, as the authors propose, or could it be a cause? It seems to me that an alternate explanation for the observed results is that the TRAP gamma- mutation leads directly (via co-regulation or other mechanisms) to large reductions in the amounts of alpha and beta, and that it is the reduction in alpha (or both alpha and beta) which causes the absence of TRAP complexes in

translocons from the gamma- fibroblasts. This alternate model has certain advantages. Alpha and beta are centrally-located, so it is quite plausible that one or both might be required for TRAP complex stability. Moreover, it becomes no longer necessary to explain away the need for gamma for stability in mammals but not in plants and algae. I conclude that the proposal that alpha is essential for stability of mammalian TRAP, while plausible, has not been conclusively demonstrated. If the authors have evidence or arguments to the contrary, I would be interested in hearing them.

Thank you for the opportunity to review this manuscript, and I congratulate the authors on a nice piece of work.

Point-by-Point Reply to Reviewers

We thank the reviewers for their thoughtful and very supportive comments. Below we address the specific points raised by the reviewers and elaborate on the corresponding changes in the manuscript.

Reviewer #1:

The manuscript by Pfeffer et al. combines a deeper analysis of the data published in 2015 in Nat. Comm. with new experiments in order to define the position of TRAP subunits in the translocon. Doing so, they are able to identify possible positions of parts of the TRAP δ and TRAP γ subunit. The so identified position of a TRAP δ domain in proximity to the OST could be an explanation for the observed underglycosylation phenotype in patients with TRAP δ mutations. All these findings are new, but only of limited general interest.

General question:

With their method, the authors nicely identify the Sec61 complex, the OST and the TRAP complex in the native translocon. However, they seem to miss other well-established constituents, like the TRAM protein or the Sec63. The authors should mention and if possible comment this problem in the discussion.

This comment refers to our previous publication (Pfeffer et al, Nature Communications 2014¹), where we clarify the core components of the ER translocon using CET and siRNA-mediated knockdown of target proteins/complexes. This approach can readily identify stoichiometric and near-stoichiometric proteins with large cytosolic or luminal domains. TRAM is the only translocon subunit candidate that has virtually no ER-luminal or cytosolic domains and mainly consists of a bundle of transmembrane helices. Hence, its position in the native translocon cannot be identified reliably using our approach. In contrast, we can rule out stoichiometric binding for several other translocon components with either large cytosolic or luminal domains (including Sec63), as discussed extensively in a recent review². The absence of Sec63 from the 'core translocon' is also in line with recent biochemical studies demonstrating that Sec62/63 only binds transiently at specific stages of the protein transport process³.

Since the reviewer comment does not directly relate to the data presented in this manuscript, we feel the discussion of other translocon components should be included in the introduction rather than in the discussion section. In the introduction, we added:

"In particular, these studies established the position of three major translocon constituents: Sec61, TRAP and the oligosaccharyl-transferase (OST) complex. Further biochemically established translocon components could not be successfully localized so far, likely because they are only transiently recruited and therefore significantly underrepresented in the average ribosome-bound translocon complex imaged in these studies."

Minor points:

The predicted orientation of the single-spanning TRAP components are more easily explained by the presence of a cleavable signal sequence. Moreover, there are biochemical data that support this prediction (e.g. the inhibition of protein transport by anti-TRAP α antibodies directed against the c-terminus of the protein).

Indeed, the presence of cleavable signal sequences confirms bioinformatic topology predictions for the single-spanning TRAP subunits. This aspect has been included in the revised results section of the manuscript:

“The presence of classical cleavable signal sequences and bioinformatic analysis based on the positive inside rule predict that three TRAP subunits (α , β , δ) □SSR1, 2, 4) consist of...”

Page 6 – at least according to the data presented in table 1 TRAP β is not more severely reduced in the TRAP γ deficient fibroblasts compared to the TRAP δ deficient ones.

Please refer to Reviewer #3, point 12.

Reviewer #2:

Pfeffer et al. have carried out a cryo-electron tomographic (CET) analysis of the organization of the translocon-associated protein (TRAP) complex. The mammalian TRAP complex is comprised of 4 subunits, whereas in plants and algae there are only two subunits. Furthermore, the TRAP gamma subunit is missing, or present in low amounts, in cells from patients with congenital disorders of glycosylation. Difference-CET analysis of the ribosome/Sec61/OST/TRAP complexes from these different systems allowed Pfeffer et al. to determine the locations of all four of the subunits of the normal mammalian TRAP complex. Their findings, summarized in Figure 5, reveal the general architecture of the ribosome/Sec61/TRAP complex. TRAP-gamma with four putative transmembrane helices, Sec61, and the lipid bilayer are clearly revealed in the tomograms. Furthermore, the tomograms indicate that TRAP-gamma is connected to the ribosome and TRAP-delta, which allows TRAP-alpha/beta to connect to Sec61 and oligosaccharyltransferase (OST).

This is important and exciting work. Although TRAP was apparent in earlier cryo-EM studies of the mammalian ribosome/Sec61 complex, its structural organization was not apparent. The present work reveals clearly for the first time the TRAP structural organization and its spatial relationship to the ribosome, Sec61, and OST. Besides being informative, I believe the work will stimulate structural studies of the TRAP subunits and thereby provide new insights to membrane protein biogenesis. A very minor point is that the meaning of OST is not clearly spelled out in the manuscript text.

We now define the abbreviation at first its use in the text.

Reviewer #3:

Manuscript Summary / Overview

In this manuscript the authors apply state-of-the-art cryo-ET imaging, subtomogram averaging, and computational sorting procedures to study translocon structure and function. By comparing algal and mutant human ribosome / translocon complexes with a previously published solution for the wild-type mammalian structure, they are able to assign presumptive locations to the TRAP delta and gamma (and by exclusion, alpha and beta) subunits. Combining these locations with information on the absence of specific subunits in some kingdoms permits inferences about the functions and mechanisms of action of the various subunits.

Key experimental observations are:

1. Bioinformatic analysis suggests that TRAP alpha, beta, and delta should contain small cytosolic domains, single transmembrane helices, and a sizeable ER luminal domain. In contrast, TRAP gamma is expected to have 4 transmembrane helices and a large cytosolic domain.
2. Homology searches indicate that TRAP gamma and delta are absent in plants and algae, while most fungi lack TRAP altogether.
3. Difference maps between human fibroblast TRAP delta- mutants and a previous wild-type mammalian structure show a single, compact region of high significance near the interface between TRAP and oligosaccharyl transferase (OST). Since Western blotting shows nearly complete elimination of TRAP delta (2% of wild type), while the other subunits are less impacted (43, 17, and 5% of wild type, respectively for alpha, beta, and gamma), the authors conclude that this region indicates the location of TRAP delta.
4. A similar attempt was made to localize TRAP gamma using a gamma- mutant. Western blotting indicated complete elimination of gamma in the mutant, but also substantial reduction of alpha, beta and delta to 17, 15, and 12% of wild type levels. Unfortunately, no TRAP complexes were found in translocons from gamma- mutants.
5. Western blots from the delta- and gamma- mutant strains indicate nearly complete elimination of the corresponding protein, but also substantial reductions in other TRAP subunits and a couple additional proteins.
6. Since algal TRAP contains alpha and beta but neither gamma nor delta subunits, the authors computed a difference map between the mammalian and *Chlamydomonas reinhardtii* complexes in another attempt to localize the gamma subunit.
 - a. OST was not well-represented in the *Chlamydomonas* samples (~14% of samples). Analysis of complexes that do contain OST indicated that algal OST lacks a large ER luminal domain found in mammals. Sequence analysis suggests that the missing region corresponds to an ~30 kDa N-terminal region of human Ribophorin II.

b. Due to the scarcity of OST containing complexes in *Chlamydomonas*, difference mapping between algal and human complexes was conducted while masking out OST. Two regions of highly significant differences were observed, one corresponding to the location previously assigned to TRAP delta. The other region is located on the cytosolic face of the endoplasmic reticulum in a position corresponding to the interface between the cytosolic domain of mammalian TRAP and 4 transmembrane helices. This is consistent with the predicted structure for TRAP gamma, so TRAP gamma is assigned to this position, spanning the membrane.

7. By a process of elimination, TRAP alpha and beta are assigned to the remaining TRAP density, a relatively compact region just below the Sec61 channel.

Based on these observations, the authors propose that:

1. In animals, TRAP gamma is essential for stability of the TRAP complex and for coordination with the ribosome / translocon complex. Gamma and delta are neither present nor required in plant and algal TRAP, so authors suggest that this need for stabilization by gamma in animals may be a relatively late evolutionary addition.

2. The location of the TRAP alpha / beta dimer just below the Sec61 channel makes it well positioned to interact with translocating peptides and to control channel conformation.

3. The delta subunit is located at the periphery of the TRAP complex, perhaps explaining the relatively minor impact of TRAP delta- mutants on stability compared to gamma- mutants. Both TRAP delta- and gamma- mutants are associated with clinically significant, congenital disorders of glycosylation, however, suggesting that delta plays an important role in coordinating TRAP and OST in mammals.

Overall Recommendation

Understanding protein processing by the translocon and the endoplasmic reticulum is an important topic of broad general interest. To the best of my knowledge, the results presented here are novel and significant, the methods used appropriate, and very well chosen and applied. Indeed, ongoing results from this group, including this manuscript, are excellent examples of how combining subtomogram averaging with computational techniques can provide insight into biologically relevant structures too large or too unstable when isolated for single particle analysis. Overall, the conclusions drawn are reasonable and well-supported by the data. Indeed, they may well be the simplest, most likely explanation for the observed results. Nevertheless, it seems to me that an alternate explanation may be possible in at least one important case, described below. Therefore, I recommend that this article be accepted for publication after editorial revision. Specifically, I suggest that the authors consider adding further discussion of alternative explanations should they and the editors agree these concerns are reasonable.

Comments, Questions, and Suggestions

In the following, I will begin with rather trivial typographical errors and formatting suggestions, proceeding to more substantive concerns.

1. Typo on page 8. Should say “wild-type” rather than “wilt-type”.

Corrected in the text.

2. Typo on page 11. Should say “Human fibroblasts” rather than “Human fibtoblasts”.

Corrected in the text.

3. Oligosaccharyl transferase should probably be fully spelled out when first introducing the OST abbreviation on page 5.

We now define the abbreviation at first use in the text.

4. In Figures 2 and 3, the “transparent red” used to flag the significantly different regions is easily confused with red color used as a key for OST in other parts of the figure. Changing either to a different color would make the figure more readily understandable.

We agree with the reviewer and have changed color for the difference densities to magenta throughout the complete manuscript.

5. Lines and / or a box might be useful to indicate the relation of Figure 2d to 2c, as was previously done for 2b and 2c. While the 45 degree rotation is clearly indicated, the change in scale was not immediately apparent, and it took me a moment to orient myself and realize what was being shown.

We added lines in order to indicate how Fig. 2d relates to Fig. 2c.

6. Also in Figure 2, why does the TRAP / OST interface show up clearly in the wild-type translocon isosurface but not as a region of significant differences? If TRAP delta is the major contributor to interaction with OST, I'd expect this region to also be missing or greatly attenuated in the TRAP delta- mutants. Is this simply a matter of the high, seven standard deviation, threshold chosen for the difference map?

The absence of visible difference density for the TRAP/OST interface in Fig. 2 results from both its low density already in the wild-type translocon and, as the reviewer suggests, the high-significance threshold used for displaying the difference density in Fig. 2.

7. Page 10 of the Discussion reads, in part, “The absence of density for transmembrane helices of the TRAP alpha/beta heterodimer (and TRAP delta) indicates that their transmembrane helices might not interact in a well-ordered fashion...”. Is it worth also commenting on and explaining the absence of detected cytosolic regions for these proteins, especially in the case of alpha which, according to the bioinformatics analysis, should have the largest of the “small” cytosolic domains?

As discussed in the manuscript, the absence of defined density for transmembrane helices of the single-spanning TRAP subunits suggests that they do not interact in a well-ordered fashion, but can move within the lipid bilayer to a certain degree. Consequently, it can be expected that the directly associated small cytosolic domains of these subunits are also rather flexibly positioned on the ER membrane and therefore averaged out in the EM density. Furthermore, the cytosolic domains of TRAP β (14 amino acids) and TRAP δ (8 amino acids) are too small to be resolved in our EM density.

In the discussion section of the manuscript, we accordingly added: "Their cytosolic domains are likely too small and flexible to be resolved, even in the higher-resolution subtomogram average."

8. Why does no TRAP gamma significant difference show up in the lumen? Is this region simply too small?

Exactly as the reviewer assumes, TRAP γ has no significantly sized luminal portions, which explains the absence of difference density in the ER lumen.

9. Page 10 of the Discussion has a parenthetical statement reading "(or only TRAP alpha in some fungi)". Given that in the Results it was previously stated that TRAP is completely missing from most fungi, would it perhaps be more clear if this parenthetical statement was changed to something like "(or only TRAP alpha in those few fungi which contain it)"?

We agree with the reviewer and rephrased the parenthetical statement as suggested.

10. In both Figures 1a and (presumably) in 5a, gray shading is used to indicate the membrane region, but this is never explicitly stated. It's pretty apparent that this is what's meant in 1a, but perhaps less so in 5a. It might also be worth indicating the transmembrane region once again in 5b (although not with gray shading!). This was already done previously in 1b, so it's doesn't seem strictly necessary, but might help to indicate relative scale and positioning between 5a and 5b.

As suggested, we labeled the ER membrane in Figs. 1a and 5a and indicated the position of the ER membrane in Fig. 5b.

The authors use Western blots to confirm protein abundance of the mutant strains microsomal fraction relative to wild type. As expected and hoped, both delta- and gamma- strains show almost complete elimination of the corresponding protein. Other TRAP and ER proteins are also affected, as indicated in Table 1. This is not surprising. Biological regulatory and functional networks are notoriously promiscuous, and feedback loops, co-regulation, and pleiotropic behavior are common in addition to effects on stability. This makes separating cause from effect challenging, leading to several questions and concerns.

11. TRAM appears to be significantly up-regulated (259% of wild type) in the delta-mutant, while Sil1 is significantly down-regulated (48% of wild type) in gamma-. Do the authors have any comments on these changes?

We extended our Western blot analysis with additional biological replicates for these two significantly changing proteins that are now included in the updated Table 1. Including the new replicates, we observe significant up-regulation for the translocating chain-associated

membrane protein (TRAM) in both patient fibroblast lines. TRAM is thought to assist Sec61 in transmembrane helix insertion ⁴ and may thus have some functional overlap with TRAP, which would explain its up-regulation. Sil1 is a nucleotide exchange factor for the ER-lumenal Hsp70 chaperone BiP. Our unpublished results suggest that Sil1 may require TRAP for biogenesis, which would explain its down-regulation in the TRAP γ and TRAP δ -deficient patient fibroblasts.

We included comments on these changes in the revised manuscript for the TRAP δ -deficient fibroblasts:

“Most other tested ER-resident proteins and complexes were not significantly affected (Table 1), except for (1) an up-regulation of the translocating chain-associated membrane protein (TRAM), which may have some functional overlap with TRAP ⁴ and could therefore partly counteract deficits in TRAP function and (2) a down-regulation of the nucleotide exchange factor Sil1, which may require TRAP for biogenesis.”

And for the TRAP γ -deficient fibroblasts:

“Consistent with the Western blot analysis of TRAP δ -deficient patient fibroblasts, most other tested ER-resident proteins and complexes were not significantly affected (Table 1), except for an up-regulation of TRAM and a down-regulation of Sil1.”

12. The statement on page 6 that “levels of TRAP alpha and TRAP beta were more severely reduced than for the previously analyzed TRAP delta-deficient fibroblasts (Fig. 3a, Table 1)” in the TRAP gamma- mutant, while literally true, seems over-stated and somewhat misleading. It unquestionably applies to alpha which is reduced to 17 \pm 4% of wild type levels in the gamma-deficient mutant versus 43 \pm 1% in delta-. For beta, however, the corresponding levels of 15 \pm 2% and 17 \pm 3% (both with N of only 2) are virtually identical.

We agree with the Reviewer that it is hard to judge whether levels of TRAP β significantly differ between the two analyzed mutations and therefore refer only to the reduction of TRAP α in the revised manuscript.

13. My greatest concern is with the observation that TRAP alpha is reduced much more dramatically in gamma- than in delta-. Is this an effect of reduced TRAP stability, as the authors propose, or could it be a cause? It seems to me that an alternate explanation for the observed results is that the TRAP gamma- mutation leads directly (via co-regulation or other mechanisms) to large reductions in the amounts of alpha and beta, and that it is the reduction in alpha (or both alpha and beta) which causes the absence of TRAP complexes in translocons from the gamma- fibroblasts. This alternate model has certain advantages. Alpha and beta are centrally-located, so it is quite plausible that one or both might be required for TRAP complex stability. Moreover, it becomes no longer necessary to explain away the need for gamma for stability in mammals but not in plants and algae. I conclude that the proposal that alpha is essential for stability of mammalian TRAP, while plausible, has not been conclusively demonstrated. If the authors have evidence or arguments to the contrary, I would be interested in hearing them.

We agree with the Reviewer that decreased TRAP complex stability in the TRAP γ mutant may not necessarily be a direct effect of the lacking TRAP γ subunit. It might indeed also result indirectly from a concomitant down-regulation of TRAP α , assuming that TRAP α is the major

complex stabilizing subunit. We rephrased the respective discussion section and included this alternative explanation in the revised manuscript:

“TRAP γ assumes a central position in the mammalian TRAP complex, binding to the ribosome on the cytosolic face of the ER membrane and coordinating the remaining TRAP subunits with the ribosome and the other translocon components. Thus, the ribosome-interacting function of TRAP γ may be required for TRAP occupancy at the translocon, explaining the complete absence of defined density for TRAP in the translocon of TRAP γ -deficient fibroblasts (Fig. 3b). The decreased stability of TRAP subunits in TRAP γ -deficient fibroblasts (Fig. 3a, Table 1) could furthermore hint at a central role of TRAP γ in assembly or stabilization of the mammalian TRAP complex. However, the ability of the TRAP α/β heterodimer to form a stable ribosome-bound subcomplex without TRAP γ in plants and algae argues against this function. Thus, an alternative explanation for the decreased TRAP complex stability in the TRAP γ -deficient fibroblasts is that it is not directly due to the lack of TRAP, but rather is caused indirectly by concomitant down-regulation of TRAP α , assuming that TRAP α is the major complex stabilizing subunit.”

Thank you for the opportunity to review this manuscript, and I congratulate the authors on a nice piece of work.

References:

- 1 Pfeffer, S. et al. Structure of the mammalian oligosaccharyl-transferase complex in the native ER protein translocon. *Nat Commun* **5**, 3072 (2014).
- 2 Pfeffer, S., Dudek, J., Zimmermann, R. & Forster, F. Organization of the native ribosome-translocon complex at the mammalian endoplasmic reticulum membrane. *Biochim Biophys Acta* **1860**, 2122-2129 (2016).
- 3 Conti, B. J., Devaraneni, P. K., Yang, Z., David, L. L. & Skach, W. R. Cotranslational stabilization of Sec62/63 within the ER Sec61 translocon is controlled by distinct substrate-driven translocation events. *Mol. Cell* **58**, 269-283 (2015).
- 4 Sauri, A., McCormick, P. J., Johnson, A. E. & Mingarro, I. Sec61alpha and TRAM are sequentially adjacent to a nascent viral membrane protein during its ER integration. *J. Mol. Biol.* **366**, 366-374 (2007).

REVIEWERS' COMMENTS:

Reviewer #1 (Remarks to the Author):

I have no further comments

Reviewer #3 (Remarks to the Author):

The authors have fully addressed all my concerns and suggestions.

Point-by-Point Reply to Reviewers

We are very pleased that the remaining two reviewers did not express any further concerns.

Reviewer #1:

I have no further comments.

Reviewer #3:

The authors have fully addressed all my concerns and suggestions.